# Spontaneous recanalization of extracranial internal carotid occlusion: A systematic scoping review

**Sarah Y. Zhang** *, **Hee Sahng Chung, Brian Dewar, Robert Fahed, Michel Shamy, Risa Shorr, Dar Dowlatshahi**

Department of Medicine (Neurology), University of Ottawa and Ottawa Hospital Research Institute, Ottawa, ON, Canada

* sazhang@toh.ca

## Abstract

### Introduction

The spontaneous recanalization of an occluded extracranial internal carotid artery (ICA) is thought to be an uncommon etiology of ischemic stroke. However, a growing number of reports describe this phenomenon. We sought to perform a scoping review of the literature to assess the prevalence of spontaneous ICA recanalization and its timing in relation to occlusion, and any patterns in imaging and treatment.

### Methods

MEDLINE, Embase, Cochrane Central Register of Controlled Trials and Web of Science were searched from inception to March 2024 for studies that included adults with spontaneous recanalization or transient occlusion of the extracranial internal carotid artery. Two investigators independently screened the studies and extracted data around recanalization proportion, timepoints, imaging, and treatment. These results were described qualitatively, and descriptive statistics were calculated where appropriate.

### Results

Of 2807 studies screened, 53 met inclusion criteria, of which 17 were cohort studies and 36 were case studies, including a total of 818 patients. The proportion of recanalization was reported in 17 cohort studies for a median of 21.2% (IQR 9.2–37.5%). Amongst the studies which reported recanalization, 46.7% of those within the cohort studies recanalized within 6 months, whereas case studies reported that 66.7% of recanalizations occurred in that same timeframe. When reported, antiplatelet treatment was the most common medical treatment pre- and post-recanalization. Doppler imaging was used to identify recanalization in 67.9% of studies, and angiography was

**Data availability statement:** All relevant data are within the manuscript and its Supporting information Files

**Funding:** The author(s) received no specific funding for this work.

**Competing interests:** The authors have declared that no competing interests exist.

used in 54.7%. Twenty-one studies reported a revascularization procedure following spontaneous recanalization.

## Conclusions

Spontaneous recanalization of an occluded extracranial carotid artery may occur, and possibly within 6 months after documented occlusion. However, clear data are lacking regarding a standard approach to imaging or treatment of patients with occluded carotid arteries.

## Introduction

Extracranial internal carotid artery (ICA) stenosis is a common cause of ischemic stroke. In individuals who experience a stroke or transient ischemic attack (TIA), the identification of ipsilateral ICA stenosis of 50–99% stenosis is an indication for urgent carotid endarterectomy (CEA) or stenting [1]. Currently, intervention is not indicated when arteries are occluded, as it is assumed that the risk for artery-to-artery embolization is reduced [2]. However, the spontaneous recanalization of a previously occluded ICA does appear to occur [3–7] and may be more frequent than previously thought. To date, there is no comprehensive summary of the literature; it is unknown how frequently these events occur, and whether they are associated with stroke recurrence warranting intervention.

Our primary objective was to perform a scoping review to synthesize and describe the existing body of literature around spontaneous recanalization after extracranial ICA occlusion. Specifically, we sought to describe the frequency of spontaneous recanalization, the imaging modalities used to identify occlusion and recanalization, the timelines of repeat imaging that detected recanalization, the risk and timing of recurrent vascular events, and any pre- and post-recanalization treatments.

## Methods

### Study protocol & registration

All supporting data and methodological detail are available within the article and online-only supplement. The protocol for this study was previous published [8] and conducted based on the guidelines of the Johana Briggs Institute (JBI) Methodology for Scoping Reviews [9] and the PRISMA extension statement for scoping reviews [10] (S1 Appendix).

### Eligibility criteria and search strategy

We included case reports/series, prospective/retrospective cohort studies, and randomized control trials of adult patients (≥18 years) presenting with spontaneously recanalized occlusion of the extracranial ICA. Stenosis and recanalization were defined as any degree of blockage ≤99%, as defined in the source studies. Additionally, all etiologies of occlusion were included, including dissection. Exclusion criteria included surgical or endovascular interventions of the carotid prior to recanalization.

Our search strategy was conducted from the date of inception to March 2024, and included the following four databases: MEDLINE, Embase, Cochrane Central Register of Controlled Trials, and Web of Science. A search strategy (S2 Appendix) was developed with the assistance of an information specialist (RS), using search terms specific to the database being searched. We only included studies published in English.

### Study selection

Screening and full-text review was conducted by two independent reviewers (SYZ, HSC) using Covidence Systematic Review software (Covidence, Melbourne, Australia). First, abstracts and titles were screened for potentially relevant articles. These articles then proceeded to full article screening using a standardized form. Disagreements in either step were resolved by consensus or with a third party.

### Data collection and synthesis of results

Data extraction was conducted independently (SYZ, HSC) using an *a priori* collection form. Publication information, study population information, and recanalization data was collected. Patient demographics included: total number of patients, primary country of recruitment, age, and sex. Recanalization data included: etiology of occlusion (i.e., dissection vs. non-dissection), timeframe of recanalization identification, pre- & post-recanalization treatment, and imaging modalities used. These results were described qualitatively. Proportion of recanalization was calculated if the paper reported on a population of both non-recanalized and recanalized cases. As per scoping review guidelines, a formal assessment of methodological quality was not performed.

## Results

### Study selection

Among the 2807 studies identified and screened, 53 met inclusion criteria (S1 Table). Reasons for exclusion are listed in Fig 1.

### Study and patient characteristics

Of the 53 included studies, 9 were prospective cohort studies, 6 were retrospective cohort studies, 2 were cohort studies with prospective data collection and retrospective data analysis, and 36 were case reports/series. In total, these studies included 818 participants (cohort n = 747, case studies n = 71). Publication year ranged from 1975 to 2023. Participant age was reported in the ICA occlusion population in 5 cohort studies and 29 case studies, with a median age of 66.2 ± 10.7 years and 54 ± 13.7, respectively (S2 Table). Thirty-five case studies reported sex in the ICA occlusion population, whereas only 6 cohort studies did so, for a median proportion of 100% and 65% male participants, respectively (S2 Table). A total of 34 studies reported the etiology of occlusion, 17 of which specified dissection as a possible etiology (9 case series, 8 cohort studies). Of these 17 dissection studies, 15 included only patients with dissection, whereas 2 studies included both patients with dissection and atheroembolic occlusions. The remaining 17 studies either did not specify an etiology, or only described atherosclerotic or cardioembolic etiologies.

### Proportion of recanalization

An overall proportion of recanalization events following carotid occlusion was calculated for 17 cohort studies, for a median of 21.2% (IQR 9.2–37.5%, S1 Table). When broken down by etiology, median proportion was 39.7% (IQR 32.8–54.3%) in dissection studies vs. 8.75% (IQR 5.70–10.5%) in non-dissection studies.

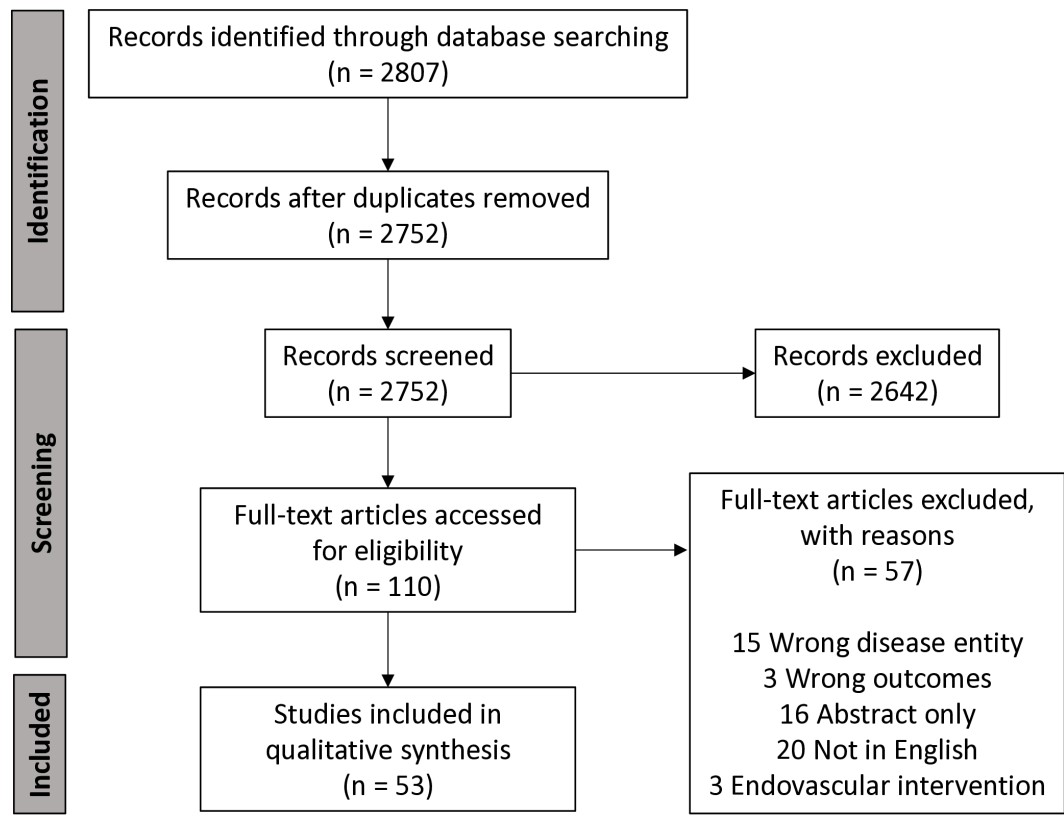

**Fig 1. PRISMA flow diagram of article review process.**

## Timeline of recanalization

Forty-nine studies (14 cohort studies, 35 case studies) reported the timing of imaging demonstrating recanalization after initial occlusion, either after symptomatic presentation (6 studies reported stroke, 3 reported TIA, 2 reported both) or through routine check-up (38 studies, asymptomatic) (Table 1). As timing of imaging demonstrating recanalization was reported differently between cases and cohort studies (i.e., individual values vs. aggregate averages), we analyzed the data separately. The median (± standard deviation) time frame in cohort studies was 7 ± 26.8 months, with a range of 5 days to 8 years, whereas case studies report a median (± standard deviation) of 3.0 ± 28.3 months, with a range of 1.5 hrs to 10 years. In the cohort studies, 20.0% of studies reported recanalization identification within the first month, 33.3% within 3 months, and 46.7% within 6 months (Fig 2A). In case studies, 33.3% of individual recanalization events were identified within the first month after initial identification of occlusion, followed by 54.5% within 3 months, and 66.7% within 6 months (Fig 2B).

## Imaging

Both occlusion and recanalization were identified through a variety of modalities, including doppler ultrasound, computed tomography angiography (CTA), magnetic resonance angiography (MRA), and conventional angiography (Table 2). In many cases, multiple modalities were used to measure both occlusion and recanalization in the same study. As such, to avoid bias and most accurately capture the full range of modalities used, we report the number of studies that used each modality. Insufficient detail was provided to report the number of patients in which each modality was used. Most studies

**Table 1. Symptoms reported at recanalization.**

| Lead author & year of publication | Symptoms at recanalization | Stroke vs. TIA |
|---|---|---|
| Saes 2007 | Amaurosis fugax to the left, aphasia, buccal angle deviation to the left and temporary worsening of motor deficit | TIA |
| Gohel 2008 | Intermittent L eye monocular blindness | TIA |
| Camporese 2011 | Two patients with non-focal symptoms, one with dysarthria, one with "transient ischemic attack" (no further details) | TIA |
| Mohammadian 2012 | One patient with an episode of right side hemiparesis and dysarthria lasting a few hours eight months later (TIA), one with right side hemiplegia and global aphasia (stroke) | TIA, stroke |
| Wopking 2013 | One patient with a short episode with paresis of the right hand and aphasia (stroke), one with (recurrent episodes with transient right-sided hemiparesis and aphasia (TIA) | TIA, stroke |
| Jin 2018 | Two patients had sudden onset of right limb weakness | Stroke |
| Wu 2018 | Two patients with recurrent stroke (no further details) | Stroke |
| Inoue 2022 | Worsened right paralysis | Stroke |
| Zhang 2023 | Transient right upper extremity weakness and worsening aphasia. | Stroke |
| Binning 2009 | Sudden loss of vision in the right eye | Stroke |
| Buslovich 2011 | Transient episodes of right-hand weakness and right amaurosis fugax | Stroke |

used doppler to identify occlusion and recanalization (62.3% and 67.9%, respectively), with conventional angiography the next most common (64.2% and 54.7%), followed by CTA (22.6%, 24.5%) and MRA (26.4%, 15.1%). The imaging modality used for identification of recanalization was not described in 3.8% of studies. There was significant heterogeneity with respect to angiographic descriptions of recanalization ranging from a higher level of detail (ex: arterial narrowing, capillary blush, and residual stenosis), minimal detail (ex: string sign), and no detail beyond "recanalization".

## Treatment and outcomes

Pre- and post-recanalization medical and surgical treatment varied between studies. At baseline (pre-recanalization), most studies reported antiplatelet, anticoagulant, or combination therapy. Among those that described baseline medical therapy, more studies reported antiplatelet-only therapy (n = 19 studies) than anticoagulant-only (n = 10), combination (n = 8), or no treatment (n = 1) (Table 3A). Post-recanalization, most studies did not describe either therapy (n = 40), though when reported, antiplatelets-only (n = 6) was more common than anticoagulant-only (n = 3), combination (n = 4), or no treatment (n = 1) (Table 3A). At the individual level, the majority of the participants' medical treatment was undescribed. However, in both pre- and post-recanalization, antiplatelet-only treatment was the most common (n = 191, n = 37, respectively).

In terms of revascularization, 47.2% of studies did not explicitly describe an intervention. Of those which did, CEA was the most common (n = 14), followed by no intervention (n = 15), stenting (n = 5), then by an undescribed intervention (n = 2) (Table 4). These interventions were completed on spontaneous recanalizations of both acute and chronic occlusions, with a time frame ranging from 3 days to 10 years post-occlusion. Out of the total 18 studies which involved an intervention, only 11 studies reported neurological outcomes. Of the 11 studies, 9 reported a "favourable" outcome in the way that there was no acute complication from surgery. Additionally, there was a lack of neurological symptoms following a specific time after intervention (1 mo – 1 yr). 2 studies report poor outcomes: one case report described rapid deterioration of neurologic condition following stent, but no further details [11]; another reported hyperperfusion syndrome following CEA [12]. In these cases, the former intervention was completed 4 days post-occlusion, while the latter was completed at 8 months.

Where revascularization was either not reported or not done, 18 studies did not describe clinical outcomes following spontaneous recanalization, 14 reported asymptomatic outcomes, 2 reported mild neurological deficits, 2 reported moderate-severe neurological deficit, and 2 reported death.

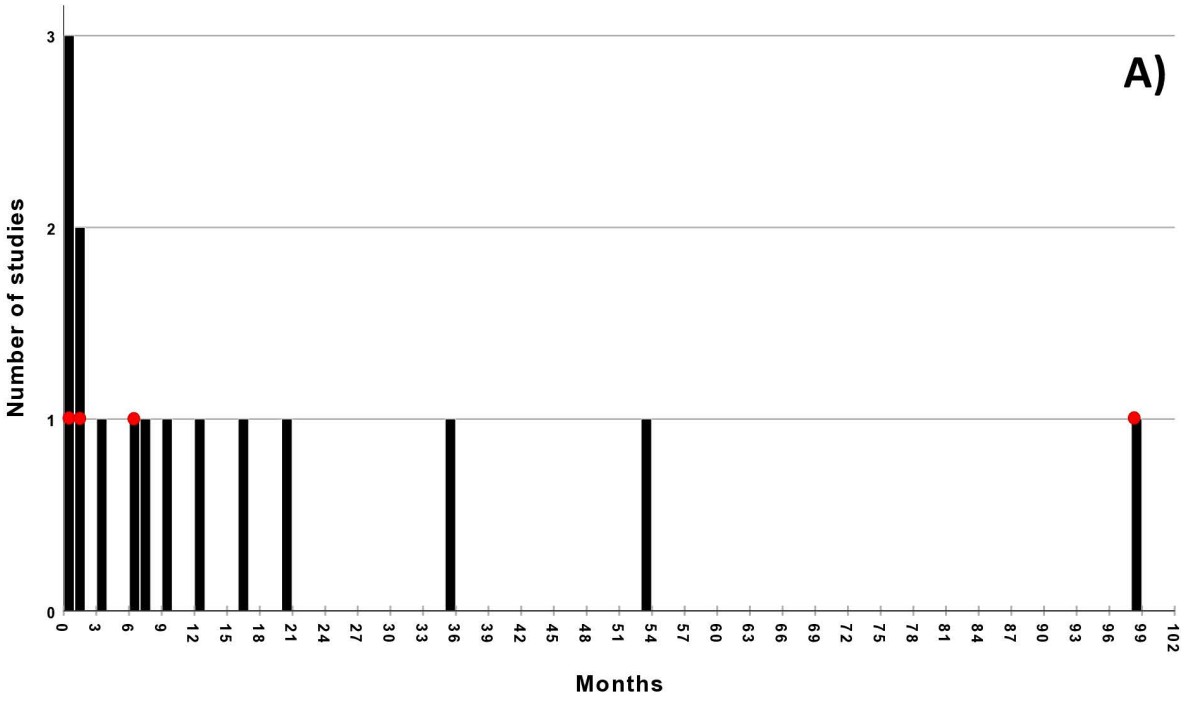

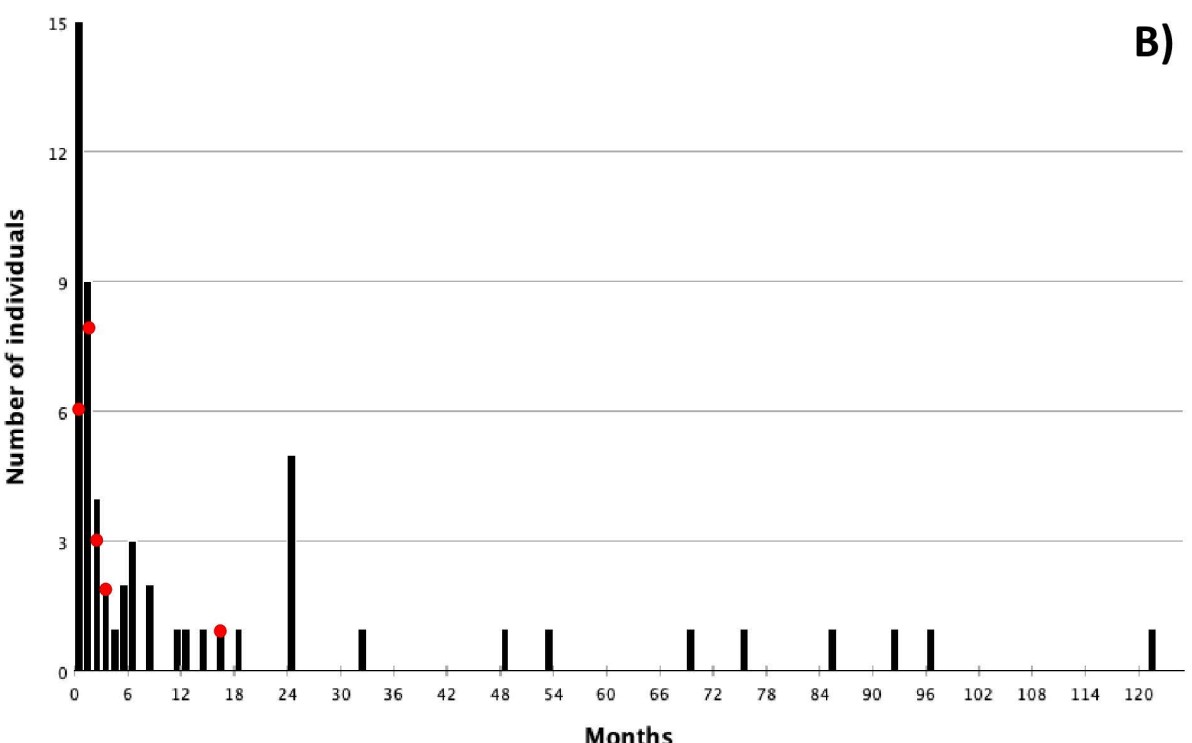

**Fig 2. Histogram of timing of imaging between occlusion and recanalization identification.** Red circles = studies including dissection. A) Distribution amongst cohort studies. B) Distribution of individual participants in case reports/series.

**Table 2. Imaging modalities used for occlusion and recanalization identification.**

| Imaging modality | Occlusion (% studies) | Recanalization (% studies) |
|---|---|---|
| Doppler | 62.3 | 67.9 |
| Angiography | 64.2 | 54.7 |
| CTA | 22.6 | 24.5 |
| MRA | 26.4 | 15.1 |
| Undescribed | 0.0 | 3.8 |

**Table 3. Pre- and post-recanalization treatments.**

**A)**

| Treatment | Pre-recanalization | Post-recanalization |
|---|---|---|
| Undescribed | 17 | 40 |
| None | 1 | 1 |
| Antiplatelet | 19 | 6 |
| Anticoagulant | 10 | 3 |
| Combo | 8 | 4 |

**B)**

| Treatment | Pre-recanalization | Post-recanalization |
|---|---|---|
| Undescribed | 557 | 768 |
| None | 1 | 1 |
| Antiplatelet | 191 | 37 |
| Anticoagulant | 56 | 8 |
| Combo | 13 | 4 |

**A)** Number of studies. **B)** Number of individuals within all studies, total n = 818.

**Table 4. Number of studies reporting surgical interventions post-recanalization.**

| Intervention | Studies (#) |
|---|---|
| CEA | 14 |
| Stenting | 5 |
| None | 15 |
| Undescribed | 25 |
| Undescribed intervention | 2 |

## Discussion

This comprehensive scoping review of spontaneous recanalization following extracranial ICA occlusion suggests that recanalization may occur in approximately 20% of cases and may be identified within 6 months after occlusion. These results suggest that recanalization does occur with sufficient frequency to be clinically relevant. However, this scoping review cannot address key questions such as whether routine surveillance is warranted, how often it should be performed, or whether revascularization should be pursued when recanalization is detected; these issues merit investigation in future prospective studies.

In qualitative literature reviews of ICA occlusion, estimates of the prevalence of recanalization are often limited to results from case studies [13,14]. In our scoping review, we show that while there is a greater absolute number of case reports/series, the study populations in the cohort studies remains high, comprising of 747 patients. However, there were

no randomized control trials (RCTs) found in our search. Despite this, from our data, we show that there may be enough patients in cohort studies to conduct a patient-level meta-analysis to better quantify prevalence and timing of recanalization. However, larger prospective studies would better inform future guideline recommendations, especially given the ambiguity surrounding timing of reimaging.

Currently, guidelines only recommend follow-up imaging and revascularization for high-grade stenotic ICAs [1], as it is thought that fully occluded arteries present a negligible risk for repeat stroke through artery-to-artery embolic events [2]. However, our results suggest that recanalization does occur following occlusion in a significant proportion of patients, thus raising the risk of delayed embolization. Given that most of the cases included in this study presented symptomatically either at occlusion or recanalization, this may present an important timepoint for intervention. However, our data suggest that there is no apparent consensus regarding the management of carotid recanalization whether symptomatic or not. Surgical or endovascular intervention may be a viable option, as there are case reports of successful revascularization following recanalization [3,12,15]. There is evidence in the literature in a shift of management for these cases. In a qualitative review by Lall et al., the authors' discussion suggest that surgical revascularization tailored to the lesion and patient would be reasonable in both asymptomatic and symptomatic recanalizations [13]. Similarly, Delgado et al. recommend intervention in symptomatic recanalizations, citing that occlusions may not be as benign as previously thought [16]. Moreover, we show that there is a non-negligible proportion of occlusions that recanalize well within 6 months, the current timeframe during which CEA is often considered following a symptomatic ischemic event [1].

This study has important limitations. First, there was significant heterogeneity in reporting and/or identifying ICA recanalization events as there is no standard timeline for surveillance imaging after carotid occlusion, raising detection bias. For this reason, our timeline data is presented as the time of identification of recanalization, rather than the time of recanalization per se. Second, the majority of recanalizations were not identified secondary to a symptomatic recurrent clinical event. For this reason, proportion and timeline findings were reported separately between case series and cohort studies. Third, it is possible that some cases reported as occlusion on non-invasive imaging may have been near occlusions with trickle flow. Additionally, given the nature of the scoping review, there is little data from the source studies regarding the exact morphology of the spontaneous recanalization, such as length of thrombus and radiological markers such as string sign. However, the majority of cases used the diagnostic gold standard of angiography to confirm occlusion in addition to non-invasive imaging, increasing the likelihood of correctly identifying an occlusion/recanalization. Finally, due to the nature of the scoping review, more rigorous statistical reporting was not implemented which limit the conclusions of this study. Overall, while this scoping review provides an important introduction to quantifying spontaneous recanalizations, a prospective cohort study with prespecified imaging protocols is ultimately needed to address the limitations of this scoping review, reduce heterogeneity and confirm these findings. Specifically, a prospective cohort study should have standardized clinical and radiological assessments with longitudinal follow-up, and utilize sophisticated imaging such as dynamic CT-angiography and MR-angiography to confirm occlusion status, describe carotid morphology following recanalization, and attempt to distinguish atheroembolic occlusion from dissection.

## Conclusion

In this study, we demonstrate that spontaneous recanalization following occlusion of extracranial ICA is possible in up to 21.2% of patients. Our findings warrant caution in the context of acute ICA occlusion and require prospective validation in a large sample using predetermined repeat imaging timepoints.

## Supporting information

**S1 Appendix. PRISMA-Scoping Review checklist.**
(PDF)

**S2 Appendix. Literature search strategy.**
(DOCX)

**S1 Table. Reference and study-level recanalization counts of 53 included studies.**
(DOCX)

**S2 Table. Demographics of 53 included studies.**
(DOCX)

## Author contributions

**Conceptualization:** Sarah Y. Zhang, Michel Shamy, Dar Dowlatshahi.

**Data curation:** Brian Dewar, Risa Shorr.

**Formal analysis:** Sarah Y. Zhang, Hee Sahng Chung.

**Methodology:** Sarah Y. Zhang, Brian Dewar, Risa Shorr, Dar Dowlatshahi.

**Project administration:** Dar Dowlatshahi.

**Supervision:** Dar Dowlatshahi.

**Writing – original draft:** Sarah Y. Zhang, Dar Dowlatshahi.

**Writing – review & editing:** Sarah Y. Zhang, Hee Sahng Chung, Brian Dewar, Robert Fahed, Michel Shamy, Dar Dowlatshahi.

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
