## [Decision Letter · Decision Letter 0]

PONE-D-24-23084Prevalence of spontaneous recanalization of extracranial internal carotid occlusion: a systematic scoping reviewPLOS ONE

Dear Dr.  Zhang,

Thank you for submitting your manuscript to PLOS ONE. After careful consideration, we feel that it has merit but does not fully meet PLOS ONE’s publication criteria as it currently stands. Therefore, we invite you to submit a revised version of the manuscript that addresses the points raised during the review process.

We look forward to receiving your revised manuscript.

Kind regards,

Atakan Orscelik

Academic Editor

PLOS ONE

Journal Requirements:

3. As required by our policy on Data Availability, please ensure your manuscript or supplementary information includes the following: A numbered table of all studies identified in the literature search, including those that were excluded from the analyses. For every excluded study, the table should list the reason(s) for exclusion. If any of the included studies are unpublished, include a link (URL) to the primary source or detailed information about how the content can be accessed. A table of all data extracted from the primary research sources for the systematic review and/or meta-analysis. The table must include the following information for each study: Name of data extractors and date of data extraction Confirmation that the study was eligible to be included in the review. All data extracted from each study for the reported systematic review and/or meta-analysis that would be needed to replicate your analyses. If data or supporting information were obtained from another source (e.g. correspondence with the author of the original research article), please provide the source of data and dates on which the data/information were obtained by your research group. If applicable for your analysis, a table showing the completed risk of bias and quality/certainty assessments for each study or outcome. Please ensure this is provided for each domain or parameter assessed. For example, if you used the Cochrane risk-of-bias tool for randomized trials, provide answers to each of the signalling questions for each study. If you used GRADE to assess certainty of evidence, provide judgements about each of the quality of evidence factor. This should be provided for each outcome. An explanation of how missing data were handled. This information can be included in the main text, supplementary information, or relevant data repository. Please note that providing these underlying data is a requirement for publication in this journal, and if these data are not provided your manuscript might be rejected.

4. We note that there is identifying data in the Supporting Information file <Zhang_SR_data export_v2.xlsx>. Due to the inclusion of these potentially identifying data, we have removed this file from your file inventory. Prior to sharing human research participant data, authors should consult with an ethics committee to ensure data are shared in accordance with participant consent and all applicable local laws. Data sharing should never compromise participant privacy. It is therefore not appropriate to publicly share personally identifiable data on human research participants. The following are examples of data that should not be shared: -Name, initials, physical address -Ages more specific than whole numbers -Internet protocol (IP) address -Specific dates (birth dates, death dates, examination dates, etc.) -Contact information such as phone number or email address -Location data -ID numbers that seem specific (long numbers, include initials, titled “Hospital ID”) rather than random (small numbers in numerical order) Data that are not directly identifying may also be inappropriate to share, as in combination they can become identifying. For example, data collected from a small group of participants, vulnerable populations, or private groups should not be shared if they involve indirect identifiers (such as sex, ethnicity, location, etc.) that may risk the identification of study participants. Additional guidance on preparing raw data for publication can be found in our Data Policy (https://journals.plos.org/plosone/s/data-availability#loc-human-research-participant-data-and-other-sensitive-data) and in the following article: http://www.bmj.com/content/340/bmj.c181.long. Please remove or anonymize all personal information (<specific identifying information in file to be removed>), ensure that the data shared are in accordance with participant consent, and re-upload a fully anonymized data set. Please note that spreadsheet columns with personal information must be removed and not hidden as all hidden columns will appear in the published file.

Reviewers' comments:

Reviewer's Responses to Questions

**Comments to the Author**

1. Is the manuscript technically sound, and do the data support the conclusions?

Reviewer #1: Partly

Reviewer #2: No

2. Has the statistical analysis been performed appropriately and rigorously? 

Reviewer #1: N/A

Reviewer #2: Yes

3. Have the authors made all data underlying the findings in their manuscript fully available?

Reviewer #1: Yes

Reviewer #2: Yes

4. Is the manuscript presented in an intelligible fashion and written in standard English?

Reviewer #1: Yes

Reviewer #2: Yes

5. Review Comments to the Author

Reviewer #1: The authors conducted a scoping review of studies reporting spontaneous recanalisation after ICA occlusion. This subject is interesting and refers to an entity that is otherwise underestimated in the literature. Though interesting the study presents some major limitations.

The main limitation is the heterogeneity regarding the included studies. Case series most often report spontaneous recanalisation in individuals who come with symptom recurrence or new events. On the other hand, prospective studies or RCTs have a more "strict" protocol of follow up imaging. Combining these disparate approaches to estimate prevalence might be confusing.

Another drawback is that it refers to ICA occlusion in general. This combines both occlusion caused by a dissection and occlusion due to atherothrombosis. These are entirely distinct entities with well documented natural history. They will,in my opinion, be handled and reported separately. More specifically, the occurrence of spontaneous recanalisation of an occlusion due to an underlying atherosclerotic plaque is a rare event for which there is a dearth of information and recommendations.

Reviewer #2: The possibility of spontaneous recanalization of an occluded internal carotid artery (ICA) has been known for years. The challenge lies not in the fact of recanalization itself but in determining the management strategy after its detection and its impact on the patient's neurological condition.

The article lacks basic data on the neurological status of patients at the time of occlusion diagnosis, the onset of neurological symptoms during the occlusion, and after the confirmation of recanalization. These are key pieces of information when discussing this topic. Please establish the neurological history of the patients analyzed.

Page 11; Line 117

The material includes acute, subacute, and chronic ICA occlusions. The mechanism of spontaneous recanalization differs across these subgroups, as do the prognosis and treatment approaches.

Do the authors have data that would allow for a separate analysis of these subgroups?

Page 12; Line 134

Imaging of extracranial ICA occlusion in the context of spontaneous recanalization requires information on the patency of this artery in the intracranial segments as well. DUS examination is performed to assess the C1 segment. In CTA, the site of restored circulation in the ICA is often not visible. The reference examination for assessing the length of ICA occlusion is selective angiography from the contralateral artery.

Do the authors have data regarding the length of ICA occlusion?

Do the authors have data on whether the spontaneous recanalization manifested as, for example, a string sign, complete recanalization, or significant stenosis in the C1 segment?

Page 13; Line 154

The surgical management of patients with spontaneous recanalization requires information on the nature of the stenosis. Did the intervention methods, such as CEA, CAS, or lack thereof, pertain to patients after acute or chronic occlusions?

What types of stenoses were they?

What were the outcomes of the surgical treatments?

Page 15; Line 207 Conclusions

On what basis did the authors conclude that patients with spontaneous recanalization who underwent surgery benefited from this approach?

6. PLOS authors have the option to publish the peer review history of their article (what does this mean? ). If published, this will include your full peer review and any attached files.

**Do you want your identity to be public for this peer review?** For information about this choice, including consent withdrawal, please see our Privacy Policy .

Reviewer #1: **Yes: ** Klearchos Psychogios

Reviewer #2: No

---

## [Author Response · Author response to Decision Letter 1]

18 Nov 2024

Comments to the Author

Reviewer #1

Comment #1: “The authors conducted a scoping review of studies reporting spontaneous recanalisation after ICA occlusion. This subject is interesting and refers to an entity that is otherwise underestimated in the literature. Though interesting the study presents some major limitations. The main limitation is the heterogeneity regarding the included studies. Case series most often report spontaneous recanalisation in individuals who come with symptom recurrence or new events. On the other hand, prospective studies or RCTs have a more "strict" protocol of follow up imaging. Combining these disparate approaches to estimate prevalence might be confusing.”

Response: We agree with the reviewer that case series may be biased towards patients with symptomatic presentations. Indeed, our study reveals that median prevalence in case series alone was 40.0%, whereas in cohort studies alone only 21.2%. For this reason, we had separated the outcomes by type of study. As discussed in our limitations, there was significant heterogeneity in reporting recanalization events, particularly amongst case series where cases were identified individually without an overarching protocol for identifying recanalizations. We have included this is in the limitations (Pg. 9-10, Lines 214-217). We have also added a sentence in our limitations section acknowledging that a prospective cohort study with prespecified imaging protocols is ultimately needed to confirm these findings and have discussed this limitation in our discussion (Pg. 10, Lines 223-226).

Comment #2: “Another drawback is that it refers to ICA occlusion in general. This combines both occlusion caused by a dissection and occlusion due to atherothrombosis. These are entirely distinct entities with well documented natural history. They will, in my opinion, be handled and reported separately. More specifically, the occurrence of spontaneous recanalization of an occlusion due to an underlying atherosclerotic plaque is a rare event for which there is a dearth of information and recommendations.

Response: The reviewer raises a valid point. We have added quantitative information about the number of dissection studies in our results section (Pg. 4, Lines 105-108). We have included the prevalence findings quantitatively in the results section (Pg. 4, Lines 114-116). Timeline results have been added qualitatively to Fig 2, and the corresponding legend has been updated (Pg. 6, Line 132).

Reviewer #2

Comment #1: “The possibility of spontaneous recanalization of an occluded internal carotid artery (ICA) has been known for years. The challenge lies not in the fact of recanalization itself but in determining the management strategy after its detection and its impact on the patient's neurological condition.”

Response: We appreciate and agree with this comment. However, we do not agree that this is necessarily a commonly held opinion amongst stroke specialists. Moreover, there is a lack of understanding within the stroke community as to the prevalence of spontaneous carotid recanalization; we wish to highlight the comment from reviewer #1 from above: “This subject is interesting and refers to an entity that is otherwise underestimated in the literature”. We believe despite the methodological limitations of a systematic scoping review, our study is a necessary first step to advance our understanding of this concept. Once we have a better understanding of the prevalence of spontaneous recanalization, we can estimate the sample sizes and inform study designs to investigate management strategies. As mentioned in our limitation section, a prospective cohort study will be the next step in confirming or refuting the findings in this scoping review (Pg. 10, Lines 223-226).

Comment #2: “The article lacks basic data on the neurological status of patients at the time of occlusion diagnosis, the onset of neurological symptoms during the occlusion, and after the confirmation of recanalization. These are key pieces of information when discussing this topic. Please establish the neurological history of the patients analyzed.”

Response: We agree with the reviewer regarding the importance of clinical information during the acute event, as it speaks to the potential for bias in patients that are symptomatic at presentation (see Reviewer #1 Comment #1). Unfortunately, a scoping review approach does not allow for a granular assessment of clinical presentation and neurological history; we were limited to the information published in the source literature. However, our intention is to use the information within this systematic review to design a prospective cohort study that will capture neurological status at initial diagnosis and at present imaging timelines.

Comment #3: “Page 11; Line 117. The material includes acute, subacute, and chronic ICA occlusions. The mechanism of spontaneous recanalization differs across these subgroups, as do the prognosis and treatment approaches. Do the authors have data that would allow for a separate analysis of these subgroups?”

Response: Unfortunately, the source data does not specify the exact mechanism of spontaneous recanalization. There is data however about the mechanism of initial occlusion, as we have outlined above in our response to Reviewer 1 (Comment #2). Etiology of occlusion will be an important piece of information that we will build into our prospective cohort study, as we have outlined in our discussion (Pg. 10, Lines 223-226).

Comment #4: “Page 12; Line 134 Imaging of extracranial ICA occlusion in the context of spontaneous recanalization requires information on the patency of this artery in the intracranial segments as well. DUS examination is performed to assess the C1 segment. In CTA, the site of restored circulation in the ICA is often not visible. The reference examination for assessing the length of ICA occlusion is selective angiography from the contralateral artery. Do the authors have data regarding the length of ICA occlusion? Do the authors have data on whether the spontaneous recanalization manifested as, for example, a string sign, complete recanalization, or significant stenosis in the C1 segment?”

Response: Unfortunately, due to the study design, we are limited to the data provided in our source material. Very few studies have information on the morphology of the recanalization, and those that report it are exclusively case series/reports. However, we have added this point to our limitation section (Pg. 10, Lines 218-221).

Comment #5” “Page 13; Line 154 The surgical management of patients with spontaneous recanalization requires information on the nature of the stenosis. Did the intervention methods, such as CEA, CAS, or lack thereof, pertain to patients after acute or chronic occlusions? What types of stenoses were they? What were the outcomes of the surgical treatments?”

Response: The timing of intervention (CEA and CAS) varied significantly amongst studies and were implemented for both acute occlusions (as soon as 3 days post-occlusion) and chronic occlusions (up to 10 yrs post-occlusion). Unfortunately, there is very limited information regarding the degree of stenosis. The majority of studies only stated “stenosis”, with occasional qualitative description (ex. “critically stenosed”) as clarification.

Not every study reported outcomes: out of 18 studies which involved an intervention, only 11 studies reported neurological outcomes. Out of these 11 studies, 9 reported a “favourable” outcome in the way that there was no acute complication from surgery. Additionally, they report that there was a lack of neurological symptoms following a specific time period after intervention (ex. ranges from 1 mo – 1 yr). Two studies described poor outcomes: one case report mentioned rapid deterioration of neurologic condition following stent, but no further details (Inoue 2022), another reported hyperperfusion syndrome following CEA (Shah 2010). We have added this data to our results section (Pg. 7, Lines 161-167). Unfortunately, due to the inherent heterogeneity of our source data set, there is no pre-established interval of follow-up. As mentioned in our revised limitations, a prospective cohort study would better address this.

Comment #6: “Page 15; Line 207 Conclusions On what basis did the authors conclude that patients with spontaneous recanalization who underwent surgery benefited from this approach?”

Response: We believe the reviewer is referring to the statement in the conclusion, “some of which may benefit from follow-up imaging and/or revascularization”. We agree this statement needs to be tempered given the limitations discussed above. We acknowledge that it is difficult to say whether revascularization is indicated and will lead to better outcomes. Our intention was to highlight the current guidelines for revascularization of significantly stenosed carotids and suggested that recanalized carotids may require a similar approach. But we take the point that one cannot necessarily extrapolate data from acute symptomatic carotids to a spontaneously recanalized carotid. We have therefore changed the conclusion to say, “The clinical significance of spontaneous recanalization is uncertain, requiring further investigation into potential benefits from follow-up imaging and/or revascularization.” (Pg. 10, Lines 230-234).

---

## [Decision Letter · Decision Letter 1]

PONE-D-24-23084R1Prevalence of spontaneous recanalization of extracranial internal carotid occlusion: a systematic scoping reviewPLOS ONE

Dear Dr. Zhang,

Thank you for submitting your manuscript to PLOS ONE. After careful consideration, we feel that it has merit but does not fully meet PLOS ONE’s publication criteria as it currently stands. Therefore, we invite you to submit a revised version of the manuscript that addresses the points raised during the review process.

We look forward to receiving your revised manuscript.

Kind regards,

Atakan Orscelik

Academic Editor

PLOS ONE

Reviewers' comments:

Reviewer's Responses to Questions

**Comments to the Author**

1. If the authors have adequately addressed your comments raised in a previous round of review and you feel that this manuscript is now acceptable for publication, you may indicate that here to bypass the “Comments to the Author” section, enter your conflict of interest statement in the “Confidential to Editor” section, and submit your "Accept" recommendation.

Reviewer #1: All comments have been addressed

Reviewer #3: All comments have been addressed

Reviewer #4: All comments have been addressed

Reviewer #5: (No Response)

2. Is the manuscript technically sound, and do the data support the conclusions?

Reviewer #1: Yes

Reviewer #3: Yes

Reviewer #4: Yes

Reviewer #5: No

3. Has the statistical analysis been performed appropriately and rigorously? 

Reviewer #1: Yes

Reviewer #3: N/A

Reviewer #4: No

Reviewer #5: No

4. Have the authors made all data underlying the findings in their manuscript fully available?

Reviewer #1: Yes

Reviewer #3: Yes

Reviewer #4: Yes

Reviewer #5: Yes

5. Is the manuscript presented in an intelligible fashion and written in standard English?

Reviewer #1: Yes

Reviewer #3: Yes

Reviewer #4: Yes

Reviewer #5: Yes

6. Review Comments to the Author

**Reviewer #1:**  All previous comments were addressed by the authors in the revised manuscript.

The manuscript has been significantly enhanced.

**Reviewer #3: ** The authors have responded to the multiple reviewer questions and concerns, particularly regarding to the limitations of the source data.

**Reviewer #4: ** This systematic scoping review of spontaneous recanalization following extracranial ICA occlusion has important limitations. The significant heterogeneity in reporting and identifying ICA recanalization events, due to the lack of a standard timeline for surveillance imaging after carotid occlusion, limits the strength of the conclusions. The majority of recanalizations were identified secondary to symptomatic recurrent clinical events, potentially biasing the results towards symptomatic cases, particularly in case reports. There is also a possibility that some cases reported as occlusion on non-invasive imaging may have been near occlusions with trickle flow. Additionally, the scoping review design provides limited data on the exact morphology of spontaneous recanalization, such as thrombus length and radiological markers. The study combines occlusions caused by dissection and atherothrombosis, which are distinct entities with different natural histories that should ideally be analyzed separately. A prospective cohort study with prespecified imaging protocols is ultimately needed to reduce heterogeneity and confirm these findings.

**Reviewer #5:**  COMMENTS:

Zhang and colleagues present a literature review to inform an important clinical question: is occluded internal carotid artery really a benign finding i.e. safe to not offer intervention as one would for a high grade stenosis? This is an interesting and valuable attempt at quantifying the prevalence of spontaneous extracranial ICA recanalization, but it has some important weaknesses that are largely addressed by the original reviewers. I note that I am not one of the original reviewers but am considering the comments of and responses to the original reviewers in my assessment. The concerns I have are really the same. These are addressed, I think, with modest effort given the apparent limitations of the underlying data, which is largely to say that the authors could do very little to address these concerns. Thus, my foremost recommendation would be to further temper the discussion of the manuscript to a level appropriate for data that is, at best, suggestive of the conclusions drawn. I acknowledge that it is not the fault of the authors that the underlying data may be of poor quality or not particularly informative for clinical use, but it is the responsibility of the authors to find diamonds in the rough that will justify their proposed prospective cohort study. There is a substantial amount of work to be done for this manuscript to be interpretable.

Major points

1. I have a hard time believing that there is no data on neurological status available in any of the manuscripts reviewed. Any information would be helpful here and would be of value to answer the fundamental question: does it matter whether an artery spontaneously recanalizes? Yes, such a patient would then meet criteria for intervention if symptomatic in a technical sense but a chronic occlusion from atherosclerotic disease versus a dynamic dissection flap is an extremely different pathology with different risk of presenting future neurologic complications. The authors at least once say most presentations for recanalization were at symptom onset with no further information provided. This is contradictory and confusing and undermines my confidence in the manuscript.

2. As a consequence of point 1), this study lacks the granularity I would expect of a scoping review that included 53 studies, even if the majority of them were case studies.

3. I do think this review article can be salvaged but it requires the addition of a section in the results section discussing the above-mentioned points, and major rewriting of discussion and restructuring of figures.

4. Consolidate figures 2 and 3 into one figure, use the additional figure space to illustrate some of the more specific data reviewers 1 and 2 had asked about.

Detailed points

Reviewer Comments

Reviewer #1

• Comment #1 Response: How can you estimate prevalence from a case series if cases are selected based on the occurrence of spontaneous recanalization? The denominator of this calculation is nebulous at best.

• Comment #2: the difference between recanalization and after occlusion due to dissection vs atherosclerotic plaque was not sufficiently discussed.

Reviewer #2

• Comment #1: Question about impact on patient’s neurological condition. This is a comment that could have and should have been addressed in more detail here and is reiterated in greater detail by this reviewer in comment #2. This is the fundamental underlying question of this work and needs to be reasoned through and answered as much as is possible with the available data.

• Comment #3 response: I appreciate the researchers including this reference to planned prospective cohort study. However, simply suggesting a possible future study is insufficient. At the very least, I would expect a detailed and candid discussion of the things that cannot be addressed in this review.

• Comment #4 response: Agree with report results of commenting on recanalization morphology in the results section, this would be interesting. Given alleged 40% prevalence from the case series evaluated, I would be very surprised that zero of them had e.g. angiographic descriptions.

• Comment #5 response: This information, regarding acuity/chronicity, is interesting and valuable and should be included in the main body of the text

• Comment #6 response: There are studies that comment on the significance of spontaneous recanalization that can and should be cited here

Methods

Lines 66-67: There must have been more exclusion criteria to lose so many (96% to be exact) of your harvested studies in the search process. If additional criteria were used, please specify these explicitly in the text.

Results

Lines 100-105: I am extremely suspicious that patient age and sex were not recorded in so many of these studies. These are very basic demographics and the lack of recording, suggest poor-quality evidence. This alone may be disqualifying for this manuscript if fundamental sociodemographics are not available from the studies.

Line 106: The meaning of "specified dissection" is not clear to me? I.e., 17 studies listed cases of dissection or 17 included it as a category regardless of whether those participants were included or not.

Lines 119-120: Please comment on the relative frequencies of symptomatic presentation vs routine check-up diagnoses.

Line 122-3: Please re-word. “Time frame in cohort studies” is total time of follow up or is time to recanalization?

Line 142: Without additional imaging modalities or could one patient have more than one imaging modality? If yes, were both counted in this review or how was decided which one was counted?

Line 158: Are these participants? Of how many total?

Lines 161-162: Lack of neurological data is again concerning re: underlying data. I think for a neurovascular pathology with intervention criteria defined by symptomatology, this again may be a disqualification of the manuscript.

Discussion

Line 179: I do not think the authors can say this likely occurs in more than 20% of cases. I think this is statement that must be tempered substantially if not entirely removed. The number of excluded studies and the opacity of the denominator in these data do not allow the authors to say this. The authors could rephrease to a “We calculate an estimated XX% incidence of recanalization based on our extremely low quality data”.

Lines 180-182: The authors must temper this statement. They are suggesting a high cost and potential intervention (which is a non-zero risk of major procedural complications up to and including stroke, MI, or death) based on low quality data.

Line 184: The majority of studies in this review were also case studies. The authors should make it clear that their estimate of prevalence is also mostly derived from case studies.

Lines 189-191: You should cite directly from the trial not just from a secondary review

Line 199: “Given that most of the cases included in this study presented symptomatically either at occlusion or recanalization…” Suggests strongly that the authors do have this data. What were the symptoms? If only from case series, that is fine but is critical and would be informative.

Line 203: “there is case report-level evidence” change to "there are case reports of successful revascularization..."

Line 220: “the majority of cases used the diagnostic gold standard of angiography“ As per Tables, the majority of patients were diagnosed via ultrasound unless multiple modalities were counted more than once

Line 229: 21.2% - What is this number derived from?

Tables

Table 2: In part A, the numbers for pre- (n=54) and post-recanalization (n=53) don't match. Where does the missing study belong?

Figures

Figure 1: Records excluded are very high. Please specify why these records were excluded.

Figure 2: This is not a good graph because there is so much overlap between datapoints and is very challenging to interpret. There is also a tremendous amount of white space in the figures. Consider alternative designs such as cumulative incidence functions.

Figure 3: Maybe I am misunderstanding but it seems that this is the same information that is presented in figure 2 and thus is totally unnecessary. If these are different, it needs to be made clearer how these are different data.

7. PLOS authors have the option to publish the peer review history of their article (what does this mean? ). If published, this will include your full peer review and any attached files.

**Do you want your identity to be public for this peer review?** For information about this choice, including consent withdrawal, please see our Privacy Policy .

Reviewer #1: **Yes: ** Klearchos Psychogios

Reviewer #3: No

Reviewer #4: **Yes: ** Robert J. Chen, MD, MPH

Reviewer #5: No

---

## [Author Response · Author response to Decision Letter 2]

21 Feb 2025

PLOSOne Reviewer Comments Round #2

Reviewer #1:

All previous comments were addressed by the authors in the revised manuscript.

The manuscript has been significantly enhanced.

Response: We thank the reviewer for their time and feedback.

Reviewer #3:

The authors have responded to the multiple reviewer questions and concerns, particularly regarding to the limitations of the source data.

Response: We thank the reviewer for their time and helpful comments.

Reviewer #4:

This systematic scoping review of spontaneous recanalization following extracranial ICA occlusion has important limitations. The significant heterogeneity in reporting and identifying ICA recanalization events, due to the lack of a standard timeline for surveillance imaging after carotid occlusion, limits the strength of the conclusions. The majority of recanalizations were identified secondary to symptomatic recurrent clinical events, potentially biasing the results towards symptomatic cases, particularly in case reports. There is also a possibility that some cases reported as occlusion on non-invasive imaging may have been near occlusions with trickle flow. Additionally, the scoping review design provides limited data on the exact morphology of spontaneous recanalization, such as thrombus length and radiological markers. The study combines occlusions caused by dissection and atherothrombosis, which are distinct entities with different natural histories that should ideally be analyzed separately. A prospective cohort study with prespecified imaging protocols is ultimately needed to reduce heterogeneity and confirm these findings.

Response: We agree fully with the reviewer. It is indeed our plan to conduct a methodologically rigorous prospective cohort study to address exactly these limitations. Our scoping review provides the justification in doing so, and we see it as our first step.

Reviewer #5

Major points

Comment #1: Zhang and colleagues present a literature review to inform an important clinical question: is occluded internal carotid artery really a benign finding i.e. safe to not offer intervention as one would for a high grade stenosis? This is an interesting and valuable attempt at quantifying the prevalence of spontaneous extracranial ICA recanalization, but it has some important weaknesses that are largely addressed by the original reviewers. I note that I am not one of the original reviewers but am considering the comments of and responses to the original reviewers in my assessment. The concerns I have are really the same. These are addressed, I think, with modest effort given the apparent limitations of the underlying data, which is largely to say that the authors could do very little to address these concerns. Thus, my foremost recommendation would be to further temper the discussion of the manuscript to a level appropriate for data that is, at best, suggestive of the conclusions drawn. I acknowledge that it is not the fault of the authors that the underlying data may be of poor quality or not particularly informative for clinical use, but it is the responsibility of the authors to find diamonds in the rough that will justify their proposed prospective cohort study. There is a substantial amount of work to be done for this manuscript to be interpretable.

Response: We completely appreciate the reviewers’ assessment of our project. We also recognize the reviewer was not one of the original reviewers so we appreciate the fresh insight and value the time taken. We hope that this scoping review will provide the foundation from which to launch a methodologically rigorous cohort study to address the limitations of this dataset. In this context, we have done our best to address the points below (within the limitations of the design) and have tempered the manuscript accordingly.

Comment #2: I have a hard time believing that there is no data on neurological status available in any of the manuscripts reviewed. Any information would be helpful here and would be of value to answer the fundamental question: does it matter whether an artery spontaneously recanalizes? Yes, such a patient would then meet criteria for intervention if symptomatic in a technical sense but a chronic occlusion from atherosclerotic disease versus a dynamic dissection flap is an extremely different pathology with different risk of presenting future neurologic complications. The authors at least once say most presentations for recanalization were at symptom onset with no further information provided. This is contradictory and confusing and undermines my confidence in the manuscript.

Response: We apologize for being unclear in our first round of responses. When we said we could not provide neurological status, we were referring to the specific presenting details such as clinical syndrome (aphasia, side of hemiplegia, NIHSS, etc.) or other physical examination findings, the details of which varied significantly from article to article. But we did not mean to imply we could not determine “symptomatic” vs “asymptomatic”; the reviewer is correct that this information was provided in some articles, and is indeed highly relevant. We have now added to the Results (under the section “Timeline of Recanalization”), “Forty-nine studies (14 cohort studies, 35 case studies) reported the timing of imaging demonstrating recanalization after initial occlusion, either after symptomatic presentation (6 studies reported stroke, 3 reported TIA, 2 reported both TIA & stroke) or through routine check-up (38 studies, asymptomatic).” Note that the previous full-text specified 42 studies which reported the timing of imaging, rather than 49. We would like to be transparent and admit this was an error in the text that was found while addressing this comment. This has no bearing on subsequent analysis as all analysis completed was on these 49 studies. However, we have now corrected the number in the text.

Comment #3: As a consequence of point 1), this study lacks the granularity I would expect of a scoping review that included 53 studies, even if the majority of them were case studies.

Response: We appreciate the reviewer’s comment. We have included more granularity in our data, including reported the number of symptomatic (TIA vs stroke) vs asymptomatic status at time of recanalization, as well as clinical status following recanalization in both revascularized and non-revascularized cases. Additional specific changes can be found in the responses to the comments below.

Comment #4: I do think this review article can be salvaged but it requires the addition of a section in the results section discussing the above-mentioned points, and major rewriting of discussion and restructuring of figures. Consolidate figures 2 and 3 into one figure, use the additional figure space to illustrate some of the more specific data reviewers 1 and 2 had asked about.

Response: We thank the reviewer for this suggestion. We have consolidated Figures 2 and 3 into one figure, and the following caption was added to the Results section (Pg. 20): “Fig 2. Histogram of timing of imaging between occlusion and recanalization identification. Red circles = studies including dissection. A) Distribution amongst cohort studies. B) Distribution of individual participants in case reports/series.”

Reviewer #5 comment regarding our response to Reviewer #1’s original comment: How can you estimate prevalence from a case series if cases are selected based on the occurrence of spontaneous recanalization? The denominator of this calculation is nebulous at best.

Response: Thank you for this important comment. Our calculations of prevalence were based only on studies which themselves reported prevalence of spontaneous recanalization over a larger sampled population of non-recanalized occlusions. This is reflected in the Method section (under Data collection and synthesis of results) which states, “Prevalence of recanalization was calculated if the paper reported on a population of both non-recanalized and recanalized cases.” Overall, there were only 3 case series which reported a prevalence. In the first case series (Irino 1975), the initial ICA occlusion population was 8 patients, 4 of which recanalized, for an overall prevalence of 50%. In Kaneda 1978, there were 5 occlusion patients, 2 of which recanalized (prevalence of 40%). Finally, in Camporese 2003, 8 of the initial 160 ICA occlusion patients recanalized, for a prevalence of 5%. The remainder of the case series/reports did not specify a larger occlusion population; therefore, they were not included in the prevalence calculation, as prevalence could not be reported.

Reviewer #5 comment regarding Reviewer #1’s comment #2: The difference between recanalization and after occlusion due to dissection vs atherosclerotic plaque was not sufficiently discussed.

Response: Thank you for bringing this to our attention. We had attempted to address this issue in our response to Reviewer #1 through the following additions: “We have added quantitative information about the number of dissection studies in our results section (Pg. 4, Lines 105-108). We have included the prevalence findings quantitatively in the results section (Pg. 4, Lines 114-116). Timeline results have been added qualitatively to Fig 2, and the corresponding legend has been updated (Pg. 6, Line 132).” We believe our study design cannot reliably distinguish between occlusions due to plaque vs dissection because we are dependent on the descriptions of authors from our source articles: we cannot assess the radiology ourselves, we cannot comment on the radiological expertise of the original authors, and we cannot measure the heterogeneity in radiological assessments between studies. While the paper would undoubtedly be strengthened by these additions, it would require access to individual patient-level data and would be outside the scope of a scoping review. In the future prospective cohort study we hope will be justified by this study, we will specifically address this issue.

Reviewer #5’s comment on reviewer #2’s comment #1: Question about impact on patient’s neurological condition. This is a comment that could have and should have been addressed in more detail here and is reiterated in greater detail by this reviewer in comment #2. This is the fundamental underlying question of this work and needs to be reasoned through and answered as much as is possible with the available data.

Response: Thank you for this comment. To offer more detail on patients’ neurological statuses, we have now added the numbers of symptomatic vs. asymptomatic status when recanalization was detected.

Reviewer #5’s comment on reviewer #2’s comment #3: I appreciate the researchers including this reference to planned prospective cohort study. However, simply suggesting a possible future study is insufficient. At the very least, I would expect a detailed and candid discussion of the things that cannot be addressed in this review.

Response: Thank you, we are happy to add specific elements that a cohort study can and should address, and we are open to suggestions from the reviewer to inform our future study design. In addition to whether an occlusion is acute, subacute or chronic, a prospective study can attempt to answer whether the etiology of an occlusion is thought to be atherosclerotic, embolic or dissection. Moreover, we can perform more sophisticated imaging to identify trickle flow and describe the morphology of the carotid after recanalization. A prospective study with standardized imaging protocols can also determine the best timelines for reimaging after a confirmed carotid occlusion. Finally, we can collect baseline demographics, clinical history and examination findings, and assess for clinical events during a follow-up period. While the details of our future cohort study are still in development (and will remain as such pending the final publication of this manuscript which will inform its design), we have now added to our limitation paragraph: “Overall, while this scoping review provides an important introduction to quantifying spontaneous recanalization, a prospective cohort study with prespecified imaging protocols is ultimately needed to address the limitations of this scoping review, reduce heterogeneity and confirm these findings. Specifically, a prospective cohort study should have standardized clinical and radiological assessments with longitudinal follow-up, and utilize sophisticated imaging such as dynamic CT-angiography and MR-angiography to confirm occlusion status, describe carotid morphology following recanalization, and attempt to distinguish atheroembolic occlusion from dissection.” Again, we are open to recommending additional components to a cohort study design at the reviewers’ suggestion. We fully intend to use this scoping review to justify the need to perform a cohort study.

Reviewer #2 comment on reviewer #2’s comment #4: Agree with report results of commenting on recanalization morphology in the results section, this would be interesting. Given alleged 40% prevalence from the case series evaluated, I would be very surprised that zero of them had e.g. angiographic descriptions.

Response: Thank you for this interesting comment and agree about the importance of including angiographic descriptions. We wish to clarify that although there are descriptions of angiographic features of the occlusions/recanalizations in the reviewed literature, we are unfortunately limited by the descriptions given in the source article. There is significant heterogeneity in level of detail of description. For example, in Irino et al.’s (1977) paper, they have multiple descriptors of the post-recanalization angiographic findings, including arterial narrowing, capillary blush, and residual stenosis. This is in contrast with Wu et al. (2018), who only described the recanalizations as a “string sign” on DSA. Finally, in studies by Kaneda et al. (1978), Adovasio et al. (2008), and Mohammadian et al. (2012), there were no descriptions of the artery besides “recanalization”. Accordingly, we have now added to the Results, “There was significant heterogeneity with respect to angiographic descriptions of recanalization ranging from a higher level of detail (ex: arterial narrowing, capillary blush, and residual stenosis), minimal detail (ex: string sign), and no detail beyond “recanalization”.”

Reviewer #5’s comment on reviewer #2’s comment #5: This information, regarding acuity/chronicity, is interesting and valuable and should be included in the main body of the text.

Response: Thank you, we agree and have added the additional information. This section (Treatment and Outcomes) now includes, “These interventions were completed on spontaneous recanalizations of both acute and chronic occlusions, with a time frame ranging from 3 days to 10 years post-occlusion” and, “In these cases, the former intervention was completed 4 days post-occlusion, while the latter was completed at 8 months” and, “Where revascularization was either not reported or not done, 18 studies did not describe clinical outcomes following spontaneous recanalization, 14 reported asymptomatic outcomes, 2 reported mild neurological deficits, 2 reported moderate-severe neurological deficit, and 2 reported death.”

Reviewer #5’s comment on reviewer #2’s comment #6: There are studies that comment on the significance of spontaneous recanalization that can and should be cited here.

Response: We thank the reviewer for their comment. We have provided further support for the significance of spontaneous recanalization and its potential indications for intervention in our discussion section: “Moreover, there is evidence in the literature in a shift of management for these cases. In a qualitative review by Lall et al., the authors’ discussion suggest that surgical revascularization tailored to the lesion and patient would be reasonable in both asymptomatic and symptomatic recanalizations (Lall et al., 2021). Similarly, Delgado et al. recommend intervention in symptomatic recanalizations, citing that occlusions may not be as benign as previously thought (Delgado et al. 2015)”. We are open to citing additional sources at the reviewers’ discretion.

Lines 66-67: There must have been more exclusion criteria to l

---

## [Decision Letter · Decision Letter 2]

PONE-D-24-23084R2Prevalence of spontaneous recanalization of extracranial internal carotid occlusion: a systematic scoping reviewPLOS ONE

Dear Dr. Zhang,

Thank you for submitting your manuscript to PLOS ONE. After careful consideration, we feel that it has merit but does not fully meet PLOS ONE’s publication criteria as it currently stands. Therefore, we invite you to submit a revised version of the manuscript that addresses the points raised during the review process.

We look forward to receiving your revised manuscript.

Kind regards,

Atakan Orscelik

Academic Editor

PLOS ONE

Journal Requirements:

Reviewers' comments:

Reviewer's Responses to Questions

**Comments to the Author**

1. If the authors have adequately addressed your comments raised in a previous round of review and you feel that this manuscript is now acceptable for publication, you may indicate that here to bypass the “Comments to the Author” section, enter your conflict of interest statement in the “Confidential to Editor” section, and submit your "Accept" recommendation.

Reviewer #4: All comments have been addressed

2. Is the manuscript technically sound, and do the data support the conclusions?

Reviewer #4: Yes

3. Has the statistical analysis been performed appropriately and rigorously? 

Reviewer #4: Yes

4. Have the authors made all data underlying the findings in their manuscript fully available?

Reviewer #4: Yes

5. Is the manuscript presented in an intelligible fashion and written in standard English?

Reviewer #4: Yes

6. Review Comments to the Author

Reviewer #4: This scoping review addresses an important clinical question; however, several specific limitations require revision. First, the heterogeneity in imaging modalities and timing of recanalization assessment significantly weakens conclusions. Clarify how multiple imaging modalities per patient were handled statistically to avoid double-counting. Second, combining dissections with atherothrombotic occlusions is problematic given distinct natural histories; separate analyses or subgroup sensitivity analyses are recommended. Third, the prevalence calculation from case series is questionable due to unclear denominators; restrict prevalence estimates to cohort studies only. Fourth, neurological outcomes and symptomatology at recanalization are inadequately detailed; explicitly report symptomatic versus asymptomatic presentations. Finally, the discussion overstates findings; temper conclusions to reflect the hypothesis-generating nature of data.

7. PLOS authors have the option to publish the peer review history of their article (what does this mean? ). If published, this will include your full peer review and any attached files.

**Do you want your identity to be public for this peer review?** For information about this choice, including consent withdrawal, please see our Privacy Policy .

Reviewer #4: **Yes: ** Robert J. Chen, MD, MPH

---

## [Author Response · Author response to Decision Letter 3]

2 Apr 2025

Reviewer #4:

This scoping review addresses an important clinical question; however, several specific limitations require revision.

Comment #1: First, the heterogeneity in imaging modalities and timing of recanalization assessment significantly weakens conclusions. Clarify how multiple imaging modalities per patient were handled statistically to avoid double-counting.

Response: Thank you for this comment. We agree that the heterogeneity of the data included in this scoping review remains high, and such emphasize that this scoping review acts as a hypothesis-generating review, with the intention of guiding a future prospective cohort study. With respect to the multiple imaging modalities. Occlusion and recanalization were often confirmed using more than one modality in the same patient or cohort (ex. occlusion initially identified with ultrasound, then confirmed with angiography or CTA/MRA). If more than one modality was used, all were quantified rather than just one; therefore, it is not a 1 modality to 1 study ratio (i.e. modalities are not mutually exclusive). As a result, the sum of the different modalities does not equal to 100%, as shown in the Imaging results section and Table 1. For example, in the paper by Kaps et al. (1990), occlusion was identified with both angiography and ultrasound doppler, therefore one study was counted towards both angiography and doppler. We opted to quantify the imaging modalities this way in order to most accurately capture the full breadth of data provided in the sample and avoid bias by choosing one modality over the other.

Comment #2: Second, combining dissections with atherothrombotic occlusions is problematic given distinct natural histories; separate analyses or subgroup sensitivity analyses are recommended.

Response: We agree with the reviewer, and have clarified and presented prevalence data separated by etiology in our results section: “When broken down by etiology, median prevalence was 39.7% (IQR 32.8 – 54.3%) in dissection studies vs. 8.75% (IQR 5.70 – 10.5%) in non-dissection studies.” (Pg. 5, lines 116-118). Additionally, we show the data of timing of imaging of recanalization in dissection vs. non-dissection studies in our updated Figure 2, where dissection data is now highlighted with red circles.

Comment #3: Third, the prevalence calculation from case series is questionable due to unclear denominators; restrict prevalence estimates to cohort studies only.

Response: We thank the reviewer for raising this point, and we have accordingly removed the prevalence data from the case series in our results section: “An overall proportion of recanalization events following carotid occlusion was calculated for 17 cohort studies, for a median prevalence of 21.2% (IQR 9.2 – 37.5%)” (Pg. 5, lines 115-116).

Comment #4: Fourth, neurological outcomes and symptomatology at recanalization are inadequately detailed; explicitly report symptomatic versus asymptomatic presentations.

Response: Thank you for this comment. We had previously attempted to address this issue in our response to Reviewer #5 in the previous round of revisions by adding the number of symptomatic vs. asymptomatic status when recanalization was detected: “Forty-nine studies reported the timing of imaging demonstrating recanalization after initial occlusion, either after symptomatic presentation (6 studies reported stroke, 3 reported TIA, 2 reported both) or through routine check-up (38 studies, asymptomatic).” Pg. 5, lines 121-123.

Comment #5: Finally, the discussion overstates findings; temper conclusions to reflect the hypothesis-generating nature of data.

Response: We have further tempered our conclusion, which now reads: “In this study, we demonstrate that spontaneous recanalization following occlusion of extracranial ICA is possible in up to 21.2% of patients. Our findings warrant caution in the context of acute ICA occlusion and require prospective validation in a large sample using predetermined repeat imaging timepoints.” We have replaced the stronger term “occurs” with “is possible” and have removed the comment about imaging and revascularization. Moreover, the abstract and discussion were previously similarly tempered in a revision.

---

## [Decision Letter · Decision Letter 3]

PONE-D-24-23084R3Prevalence of spontaneous recanalization of extracranial internal carotid occlusion: a systematic scoping reviewPLOS ONE

Dear Dr. Zhang,

Thank you for submitting your manuscript to PLOS ONE. After careful consideration, we feel that it has merit but does not fully meet PLOS ONE’s publication criteria as it currently stands. Therefore, we invite you to submit a revised version of the manuscript that addresses the points raised during the review process.

We look forward to receiving your revised manuscript.

Kind regards,

Atakan Orscelik

Academic Editor

PLOS ONE

Journal Requirements:

Reviewers' comments:

Reviewer's Responses to Questions

**Comments to the Author**

1. If the authors have adequately addressed your comments raised in a previous round of review and you feel that this manuscript is now acceptable for publication, you may indicate that here to bypass the “Comments to the Author” section, enter your conflict of interest statement in the “Confidential to Editor” section, and submit your "Accept" recommendation.

Reviewer #4: All comments have been addressed

2. Is the manuscript technically sound, and do the data support the conclusions?

Reviewer #4: Yes

3. Has the statistical analysis been performed appropriately and rigorously? 

Reviewer #4: Yes

4. Have the authors made all data underlying the findings in their manuscript fully available?

Reviewer #4: Yes

5. Is the manuscript presented in an intelligible fashion and written in standard English?

Reviewer #4: Yes

6. Review Comments to the Author

Reviewer #4: The authors have made commendable revisions in response to prior concerns; however, some issues persist. While imaging heterogeneity is acknowledged, the handling of multiple modalities per patient remains insufficiently clarified—quantifying per study rather than per patient risks duplication bias. Although dissection and non-dissection etiologies are now separated, the categorization could benefit from consistent application and clearer definitions across analyses. Prevalence estimates have appropriately excluded case series; this correction improves methodological rigor. Neurological outcomes and symptom status at recanalization are partially addressed but remain underreported—explicit patient-level counts and clinical context are needed. Finally, despite some moderation, elements of the discussion continue to imply clinical applicability beyond the scope of this review; further tempering is warranted. Overall, the manuscript addresses an important gap, but further revision is necessary to ensure analytical precision and appropriate interpretation of results.

7. PLOS authors have the option to publish the peer review history of their article (what does this mean? ). If published, this will include your full peer review and any attached files.

**Do you want your identity to be public for this peer review?** For information about this choice, including consent withdrawal, please see our Privacy Policy .

Reviewer #4: **Yes: ** Robert J. Chen, MD, MPH

---

## [Author Response · Author response to Decision Letter 4]

12 May 2025

PLOS One Reviewer Comments Round #4

Reviewers #1 & 3: responses were accepted and no further questions as of January 20th, 2025.

Reviewer #5: no further questions as of March 24th, 2025.

Reviewer #4 Dr. Robert J. Chen (March 24th, 2025):

The authors have made commendable revisions in response to prior concerns; however, some issues persist.

Comment #1: While imaging heterogeneity is acknowledged, the handling of multiple modalities per patient remains insufficiently clarified—quantifying per study rather than per patient risks duplication bias.

Response: We have added information in the results section clarifying how the various modalities were quantified at the study level: “In some cases, multiple modalities were used to measure both occlusion and recanalization in the same study. As such, to avoid bias and most accurately capture the full range of modalities used, we report the number of studies that used each modality. Insufficient detail was provided to report the number of patients in which each modality was used.” (Pg. 7, lines 140-143).

Unfortunately, not all source studies provided sufficient detail on which modalities were used in which individuals, specifically in cohort studies. Four out of 17 cohort studies did not specify the imaging modality at the individual patient level (Damania 2016, Delgado 2015, Morris-Stiff 2015, and Paciaroni 2005); therefore, we are limited to quantifying at the study level. While we completely appreciate why Dr. Chen is asking for this information, these data require patient-level meta-analysis that is beyond the scope of this study.

Comment #2: Although dissection and non-dissection etiologies are now separated, the categorization could benefit from consistent application and clearer definitions across analyses.

Response: First and foremost, we are extremely appreciative of the time and effort reviewer Dr. Chen has invested in improving our manuscript. We also understand that Dr. Chen has identified an important point about dissections that we have incorporated into the design of a prospective study that is informed by this systematic review: namely, that the definition of dissection needs to be standardized a priori. But for this study, after careful consideration and consultation with our Ottawa Method Centre colleagues, we feel we are now at risk of straying too far from our original published protocol to attempt to impose a standardized definition of dissection on already published work. Moreover, while the intention is to provide clarity on distinct etiological causes of occlusion, we do not believe that further clarification would be beneficial to the average reader.

Comment #3: Prevalence estimates have appropriately excluded case series; this correction improves methodological rigor.

Response: We thank the reviewer for their comment.

Comment #4: Neurological outcomes and symptom status at recanalization are partially addressed but remain underreported—explicit patient-level counts and clinical context are needed.

Response: Thank you for this comment. Unfortunately, a patient-level pooled analysis is beyond the scope of this paper. As a result, we do not have patient level data about symptoms and outcomes. Nevertheless, we have added more detail to the instances where neurological deficits were described after spontaneous recanalization occurred in a new table (see below) and added this table to our results section: “Forty-nine studies (14 cohort studies, 35 case studies) reported the timing of imaging demonstrating recanalization after initial occlusion, either after symptomatic presentation (6 studies reported stroke, 3 reported TIA, 2 reported both) or through routine follow up (38 studies, asymptomatic) (Table 1).” (Pg. 6). Because this is not a patient-level meta-analysis, we are limited in our descriptions to what the source material provided.

[Due to formatting issues, please see attached Reviewer's comments and manuscript files for Table 1]

Comment #5: Finally, despite some moderation, elements of the discussion continue to imply clinical applicability beyond the scope of this review; further tempering is warranted.

Response: Thank you for your comment. This review is only intended to inform a prospective study, and we did not intend to imply that imply its results should dictate clinical management. We have searched carefully through the manuscript and removed an additional statement from the abstract, “whether intervention is needed, and when any intervention should occur”.

We do not believe the introduction makes management claims. Neither do the methods and results.

In the discussion, we have reworded the opening paragraph to the following: “These results suggest that recanalization does occur with sufficient frequency to be clinically relevant. However, this scoping review cannot address key questions such as whether routine surveillance is warranted, how often it should be performed, or whether revascularization should be pursued when recanalization is detected; these issues merit investigation in future prospective studies.”

The third paragraph summarizes the matter of revascularization as published in the literature – this is important because it contextualizes our study. We feel that omitting this section would be doing a disservice to our readers. Nowhere in this paragraph do we suggest a management change. Rather, we again highlight the timeline of possible recanalization (see Results section: Timeline of recanalization, Pg. 5).

We then conclude with a neutral data statement and an appeal for validation.

We wish to emphasize that we agree with Dr. Chen: we are not trying to imply clinical applicability, and if there are any parts of our manuscript that say otherwise, we are open to revising them accordingly. For this reason, we are open to Dr. Chen suggesting specific textual edits to further temper the text if required; in this scenario, we will formally acknowledge Dr. Chen’s contributions in a revision.

Comment #6: Overall, the manuscript addresses an important gap, but further revision is necessary to ensure analytical precision and appropriate interpretation of results.

Response: Thank you for this comment and we have endeavoured to address the comments raised above.

---

## [Decision Letter · Decision Letter 4]

PONE-D-24-23084R4Prevalence of spontaneous recanalization of extracranial internal carotid occlusion: a systematic scoping reviewPLOS ONE

Dear Dr. Zhang,

Thank you for submitting your manuscript to PLOS ONE. After careful consideration, we feel that it has merit but does not fully meet PLOS ONE’s publication criteria as it currently stands. Therefore, we invite you to submit a revised version of the manuscript that addresses the points raised during the review process.

We look forward to receiving your revised manuscript.

Kind regards,

Atakan Orscelik

Academic Editor

PLOS ONE

Journal Requirements:

Reviewers' comments:

Reviewer's Responses to Questions

**Comments to the Author**

1. If the authors have adequately addressed your comments raised in a previous round of review and you feel that this manuscript is now acceptable for publication, you may indicate that here to bypass the “Comments to the Author” section, enter your conflict of interest statement in the “Confidential to Editor” section, and submit your "Accept" recommendation.

Reviewer #4: All comments have been addressed

2. Is the manuscript technically sound, and do the data support the conclusions?

Reviewer #4: Partly

3. Has the statistical analysis been performed appropriately and rigorously? 

Reviewer #4: Yes

4. Have the authors made all data underlying the findings in their manuscript fully available?

Reviewer #4: Yes

5. Is the manuscript presented in an intelligible fashion and written in standard English?

Reviewer #4: Yes

6. Review Comments to the Author

Reviewer #4: This scoping review addresses an important topic but has several methodological and interpretive limitations. The authors pool heterogeneous studies (cohorts and case reports) without adequate weighting or bias assessment, making the reported “prevalence” (median ~21%) potentially misleading. Inclusion criteria and definitions (true occlusion vs high-grade stenosis, and recanalization threshold) should be clarified. Different imaging modalities (Doppler US, CTA, angiography) and variable follow-up intervals are not accounted for, raising detection bias. Statistical reporting relies on unweighted descriptive metrics without confidence intervals or heterogeneity analysis. There is no appraisal of study quality or publication bias. Case reports inherently bias timing and frequency; the claim that recanalization is more common than thought seems overstated. Study-level data (sample sizes, recanalization counts) should be reported, and a meta-analytic or weighted approach considered to provide confidence intervals. Overall conclusions must be tempered given these limitations.

7. PLOS authors have the option to publish the peer review history of their article (what does this mean? ). If published, this will include your full peer review and any attached files.

**Do you want your identity to be public for this peer review?** For information about this choice, including consent withdrawal, please see our Privacy Policy .

Reviewer #4: **Yes: ** Robert J. Chen, MD, MPH

---

## [Author Response · Author response to Decision Letter 5]

22 May 2025

Reviewers #1 & 3: responses were accepted and no further questions as of January 20th, 2025.

Reviewer #5: no further questions as of March 24th, 2025.

Journal Requirements: Please review your reference list to ensure that it is complete and correct. If you have cited papers that have been retracted, please include the rationale for doing so in the manuscript text, or remove these references and replace them with relevant current references. Any changes to the reference list should be mentioned in the rebuttal letter that accompanies your revised manuscript. If you need to cite a retracted article, indicate the article’s retracted status in the References list and also include a citation and full reference for the retraction notice.

Response: We have gone through our bibliography and all included references and have not found evidence of article retraction.

Reviewer #4 Dr. Robert J. Chen (March 24th, 2025):

This scoping review addresses an important topic but has several methodological and interpretive limitations.

Comment #1: The authors pool heterogeneous studies (cohorts and case reports) without adequate weighting or bias assessment, making the reported “prevalence” (median ~21%) potentially misleading.

Response: We have removed the word prevalence and replaced it with “proportion”. We have also removed it from the title.

Comment #2: Inclusion criteria and definitions (true occlusion vs high-grade stenosis, and recanalization threshold) should be clarified.

Response: We have included these clarifications in our Methods section: “Stenosis and recanalization were defined as any degree of blockage ≤99%, as defined in the source studies. Additionally, all etiologies of occlusion were included, including dissection.” (Pg. 3-4, lines 72-74).

Comment #3: Different imaging modalities (Doppler US, CTA, angiography) and variable follow-up intervals are not accounted for, raising detection bias. Statistical reporting relies on unweighted descriptive metrics without confidence intervals or heterogeneity analysis. There is no appraisal of study quality or publication bias. Case reports inherently bias timing and frequency; the claim that recanalization is more common than thought seems overstated. Study-level data (sample sizes, recanalization counts) should be reported, and a meta-analytic or weighted approach considered to provide confidence intervals. Overall conclusions must be tempered given these limitations.

Response: We thank the reviewer for raising these limitations. Our Ottawa Methods Centre (internationally recognized in their work with systematic reviews) have advised that heterogeneity analysis or appraisal of study quality are not required for scoping reviews and were therefore not included in the published protocol. In response to the suggestions, we have provided a supplementary table with the additional information requested above (for the sake of formatting, please refer to the Response to Reviewers and manuscript files for the full table). Moreover, we further tempered our abstract and discussion:

Abstract: “Spontaneous recanalization of an occluded extracranial carotid artery may occur, and possibly within 6 months after documented occlusion. However, clear data are lacking regarding a standard approach to imaging or treatment of patients with occluded carotid arteries.” (Pg. 2, Lines 34-36).

Discussion/limitations:

“These results suggest that recanalization does occur with sufficient frequency to be clinically relevant. However, this scoping review cannot address key questions such as whether routine surveillance is warranted, how often it should be performed, or whether revascularization should be pursued when recanalization is detected; these issues merit investigation in future prospective studies.” (Pg. 10, lines 217-221)

“Second, the majority of recanalizations were not identified secondary to a symptomatic recurrent clinical event. For this reason, proportion and timeline findings were reported separately between case series and cohort studies.” (Pg. 11, lines 259-260).

“Finally, due to the nature of the scoping review, more rigorous statistical reporting was not implemented which limit the conclusions of this study.” (Pg. 11, lines 267-269).

---

## [Editor Report · Decision Letter 5]

Spontaneous recanalization of extracranial internal carotid occlusion: a systematic scoping review

PONE-D-24-23084R5

Dear Dr. Zhang,

We’re pleased to inform you that your manuscript has been judged scientifically suitable for publication and will be formally accepted for publication once it meets all outstanding technical requirements.

Kind regards,

Atakan Orscelik

Academic Editor

PLOS ONE
---

## [Editor Report · Acceptance letter]

PONE-D-24-23084R5

PLOS ONE

Dear Dr. Zhang,

I'm pleased to inform you that your manuscript has been deemed suitable for publication in PLOS ONE. Congratulations! Your manuscript is now being handed over to our production team.

Kind regards,

on behalf of

Dr. Atakan Orscelik

Academic Editor

PLOS ONE